# miRNAs and Alcohol-Related Hepatitis

**DOI:** 10.3390/cimb47121048

**Published:** 2025-12-15

**Authors:** Dinuka Bandara, Clara Ashraf Boshra Shaker Romany, Vikash Kumar, Aalam Sohal, Mohanad Al-Qaisi, Nilofar Najafian

**Affiliations:** 1Department of Internal Medicine, Creighton University, Phoenix, AZ 85012, USA; dinukabandara@creighton.edu (D.B.); clararomany@creighton.edu (C.A.B.S.R.); 2Division of Gastroenterology and Hepatology, Creighton University, Phoenix, AZ 85012, USA; vikashkumar@creighton.edu (V.K.); mohananalqaisi@creighton.edu (M.A.-Q.); nilofar.najafian@commonspirit.org (N.N.)

**Keywords:** miRNA, alcohol-related hepatitis, alcoholic hepatitis, biomarkers, pro-inflammatory, NF-κB activation, extracellular vesicles, stellate cell activation

## Abstract

Alcohol-related hepatitis (AH) is a severe, life-threatening liver inflammation caused by chronic heavy drinking, with high short-term mortality despite abstinence and supportive care. The pathophysiology involves a compromised gut–liver axis, activation of Kupffer cells, stimulation of hepatic stellate cells, and progressive fibrosis. Increasing evidence suggests that microRNAs (miRNAs), small non-coding RNAs that regulate gene expression post-transcriptionally, play a role as modulators of these processes. Understanding dysregulated miRNAs in AH may provide insights into novel diagnostic and therapeutic interventions. Several miRNAs have been identified as critical regulators of AH pathogenesis. Upregulated miRNAs, including miRNA-217, miRNA-182, let-7b, miRNA-21, and miRNA-34a, promote inflammation through NF-κB activation, Toll-like receptor (TLR) signaling, cytokine production, and ductular reactions. Conversely, downregulated miRNAs such as miRNA-148a, miRNA-30e, and miRNA-483-3p are associated with impaired hepatocyte differentiation, dysregulated oxidative stress responses, and enhanced Mallory–Denk body formation. Considering that miRNAs are pivotal regulators of AH pathophysiology including immune activation, hepatocyte death, fibrosis, and metabolic dysregulation, their altered expression patterns not only illuminate key pathogenic pathways but also provide promising avenues for biomarker discovery and therapeutic targeting. This review aims to summarize the current literature regarding the miRNA profiles involved in alcohol-related hepatitis, their individual mechanistic roles in pathogenesis of AH, and their potential for biomarkers.

## 1. Introduction

Alcoholic-associated liver disease (ALD) refers to a spectrum of liver disorders caused by chronic excessive alcohol intake, ranging from mild steatosis to alcoholic hepatitis (AH), fibrosis, cirrhosis, and hepatocellular carcinoma [1]. Of these, alcohol-related hepatitis is an acute inflammatory phenotype of ALD that is severe and sometimes life-threatening, with characteristics of acute hepatocellular injury, significant hepatic inflammation, and progressive fibrosis [2]. Around 20–40% of patients with alcohol abuse are estimated to develop alcohol-related hepatitis at some point in their lives, and the condition itself is accountable for a high proportion of alcohol-related morbidity and mortality worldwide [2]. Despite abstinence and other supportive therapies, 30-day mortality from severe alcoholic hepatitis remains high at 30–50% [3]. This highlights the need for a more comprehensive understanding of underlying physiologic processes to identify potential treatment modalities [3].

The current understanding of alcohol-related hepatitis pathophysiology is complex, driven by a compromised gut–liver axis that leads to dysbiosis, activation of Kupffer and hepatic stellate cells, release of pro-inflammatory cytokines, perpetuation of liver injury, and fibrogenesis [4]. Emerging evidence has identified microRNAs (miRNAs) as important post-transcriptional regulators in various steps of alcoholic hepatitis development [5]. miRNAs are small endogenous non-coding RNAs that control gene expression at the post-transcriptional level by recognizing complementary sequences within the 3′ Untranslated Region (UTR) of targeted mRNAs [6]. Several miRNAs have been shown to target key pathways involved in alcoholic hepatitis pathogenesis, including the stimulation of hepatic stellate cells, cytokine signaling, and programmed cell death [5]. It has been further observed that alcohol exposure alters circulating and hepatic miRNA profiles [7].

New evidence highlights that identifying dysregulated miRNAs may be beneficial for novel diagnostic and therapeutic interventions in alcohol-related hepatitis [7,8]. Deciphering this interplay is therefore crucial for determining the specific active miRNAs involved in pathological contexts, thereby enabling the design of novel therapeutic modalities for this high-risk population. This review article summarizes the current understanding regarding the involvement of miRNAs in alcohol-related hepatitis pathogenesis, as well as their translational implications.

## 2. Search Methodology

This paper systematically reviewed a comprehensive literature search which was conducted through PubMed, Google Library, and Cochrane Library to identify studies relevant to dysregulated miRNA profiles in the setting of alcohol-related hepatitis. The search was performed between 17 June 2025–25 August 2025 using a combination of keywords and controlled vocabulary terms. Boolean operators (“and”, “or”) were used to combine search terms. The search was limited to the English language and a date range between 2002–present. Study types included human, animal models, in vitro mechanistic studies, and reviews. Search results were screened based on pre-defined inclusion criteria (including original data on miRNA expression, association with alcohol-related hepatitis, in vitro/in vivo studies, etc.) and exclusion criteria (including studies that do not delineate alcohol-related liver disease from other pathologies). See attached PRISMA diagram in Figure 1.

## 3. TLR, NF-kB Signaling, and Pro-Inflammatory Cytokine Production

One of the key mediators in the pathological spectrum of alcoholic hepatitis is the production of pro-inflammatory cytokines (TNF-α, IL-1, IL-6, IL-8, IL-17, etc.). These cytokines, primarily produced by Kupffer cells, ultimately contribute to progressive liver damage by driving the inflammatory response [10]. Kupffer cells are the primary type of macrophage that resides in the liver, lining the hepatic sinusoids. These cells express receptors, such as Toll-like receptor 4 (TLR4) that recognize bacterial endotoxins, leading to the mass activation and release of the aforementioned inflammatory cytokines [11]. Recent studies have demonstrated that alcoholic liver disease is associated with alterations in the gut microbiome and disruption of the gut–liver axis, leading to translocation of gut-derived lipopolysaccharide (LPS) to the liver [12]. LPS enters the liver via the portal vein and activates Kupffer cells by binding to TLR4, stimulating the production of the above pro-inflammatory cytokines [13]. Ethanol also sensitizes Kupffer cells, enhancing LPS-stimulated release of inflammatory cytokines [13]. Therefore, both ethanol and LPS have been shown to contribute to the progression of alcoholic hepatitis synergistically. See Figure 2 for a list of the deregulated miRNAs in AH.

### 3.1. miRNA-217

Among the mediators of this progression of alcohol-related hepatitis is miRNA-217, which was already known to aggravate ethanol-induced steatosis in hepatocytes. In 2015, Yin et al. demonstrated that LPS and ethanol each significantly up-regulated miRNA-217 in cultured RAW 264.7 macrophages and primary Kupffer cells in vitro [13]. They also observed that ethanol and LPS stimulation increased the expression of inflammatory genes, including TNF-α and IL-6, in Kupffer cells, with the highest levels observed in Kupffer cells from ethanol-fed mice [13]. In vitro, ethanol exposure further boosted LPS-induced IL-6 significantly [13]. In RAW 264.7 macrophages, miRNA-217 overexpression was found to raise the mRNA levels of IL-1b, IFN-γ, MCP-1, and iNOS [13]. miRNA-217 overexpression amplified ethanol + LPS induced expression of IL-1b, MCP-1, and iNOS [13]. Collectively, these results suggest that miRNA-217 modulates the expression of a cluster of pro-inflammatory cytokines in in vitro macrophages exposed to ethanol, LPS, or a combination of ethanol and LPS. Furthermore, ethanol + LPS mediated expression of miRNA-217 in macrophages functionally attenuates expression of SIRT1, an inhibitor of NF-kB (a crucial transcription factor in the pathogenesis of inflammation) and nuclear factor of activated T-cells 4 (NFATc4) activity [13] (see Figure 3). Given the demonstrated influence of miRNA-217 in these pathways suggest a potential avenue for therapeutic research involving knockdown of this specific miRNA to mitigate the inflammatory response in AH.

### 3.2. Let-7b

Massey et al. aimed to understand the role of TLR-7 expression in the setting of alcoholic hepatitis as it relates to endogenous miRNA activation [14]. TLR-7 had been previously shown to have increased expression in the setting of chronic alcohol exposure in mice [15] and in humans with end-stage alcohol-related liver disease [16], but its functional effect had not been previously explored. They were able to demonstrate that ethanol increased expression of MyD88 and NFkB-p65, and additionally induced the release of a specific miRNA known as let-7b in hepatocyte-derived microvesicles. Taken together, these findings suggest that ethanol increases TLR-7 signaling through the induction of the TLR-7 receptor and its signaling components (MyD88, NF-ΚB-P65) and through an increased release of let-7. The study also explored the association of TLR-7 with human alcoholic hepatitis using RNA sequencing data of human liver tissue and found that mRNA expression of TLR-7 was increased by more than 300% in patients with alcoholic hepatitis [14]. RNA sequencing using biopsy samples from patients with AH (*n* = 29) or NASH (*n* = 9) as well as fragments from normal liver tissue (*n* = 10) also showed that six different let-7 microRNA variants were increased in patients with alcohol-related hepatitis, including let-7b [14]. This meant that TLR-7 expression and let-7b expression were significantly increased in alcohol-related hepatitis and positively correlated with the downstream inflammatory components of MyD88/NFkB-p65 activation, including pro-inflammatory cytokines (e.g., IL-8, TNF-α, COX2, CXCL1, TIMP1) and NEAT1 (a positive regulator of IL-8 expression) [14] (see Figure 4).

### 3.3. miRNA-182

In a comprehensive study conducted by Blaya et al., miRNA profiling suggested that alcohol-related hepatitis expressed a profile of miRNAs that was distinctive from other chronic liver diseases, including alcohol related liver disease [17]. A multidimensional scaling analysis further identified miRNA-182 to be the most highly expressed serum miRNA in alcohol-related hepatitis [17]. Its expression level correlated strongly with MELD scores [17]. Moreover, a positive correlation was found between the expression of miRNA-182 and serum bilirubin levels [17]. The study also uncovered an association between miRNA-182 expression and keratin 7 and epithelial cell adhesion molecule (EPCAM), two markers of ductular reaction whereby there is expansion of the small bile ductules within the liver’s portal areas as a direct response to cholangiocyte proliferation or the transdifferentiation of hepatocytes into cholangiocytes [18]. In their cohort of patients who were admitted with alcohol-related hepatitis, miRNA-182 expression was also higher in those patients who died within 90 days of admission [17]. Altogether, this signified that the expression levels of miRNA-182 correlated strongly with disease severity, extent of ductular reaction, and short-term mortality, suggesting its potential as a prognostic biomarker in the pathogenesis of alcohol-related hepatitis.

Blaya et al. further evaluated the role of miRNA-182 in liver injury in general by using miRNA-182 decoys to block miRNA-182 in mouse models fed with a 3,5-diethoxycarbonyl-1,4-dihydrocollidine (DDC) diet, an in vivo model diet which mimics cholestatic disease through periductular fibrosis [17]. These miRNA-182 decoys were constructs that acted as sponges, binding to mature miRNAs and preventing them from pairing with target mRNA. Administration of these decoys in DDC-treated mice induced a reduction in liver injury as evidenced by a decrease in ALT, AST, and LDH levels compared with decoy-control mice [17]. Interestingly, the decoy-treated mice exhibited decreased total bile acids in liver tissue, suggesting that miRNA-182 may be involved in bile acid metabolism and cholestasis [17]. As bile acid accumulation can have a cytotoxic effect on the liver, it is reasonable that the reduction in liver injury in decoy-treated mice may be an indirect effect of reduced bile acid levels. However it is important to acknowledge that differences exist in bile acid metabolism across different species, including differences in the primary bile acid composition and bile acid conjugation. Furthermore, it was demonstrated that several key inflammatory genes (Mcp-1, Ccl20, Cxcl5, and Cxcl1) and the anti-apoptotic gene Bcl-2 were reduced in DDC-treated mice that received the decoy treatment [17]. This suggests that inhibition of miRNA-182 could be a component in attenuating the inflammatory response in liver disease.

### 3.4. miRNA-21

Another aberrantly expressed miRNA in alcoholic liver injury is miRNA-21, a known mediator of IL-6/Stat3 signaling and subsequent inflammatory activities associated with the liver [19]. Human stellate cells (HSCs), which reside in the perisinusoidal space between sinusoidal endothelial cells and hepatocytes, integrate cytokine-mediated inflammatory responses in the sinusoids themselves and transmit them to the liver parenchyma [20]. As it was previously known that several miRNAs are upregulated in HSCs in the setting of alcoholic liver disease, Wu et al. set out to investigate the role of miRNA-21 in the development of alcohol-related hepatitis. Given that miRNAs generally inhibit gene expression, the study focused on upregulated genes in HSCs transfected with anti-miRNA-21. PCR array analysis showed that anti-miRNA-21 upregulated Von Hippel-Lindau (VHL) mRNA expression [21]. Using real-time PCR and ELISA confirmation, it was discovered that the overexpression of VHL significantly decreased the expression levels of IL-6, MCP-1, and IL-1β mRNAs [21]. Altogether, this highlighted how the activation of VHL expression by miRNA-21 knockout and anti-mRNA-21 suppressed the production of pro-inflammatory cytokines in human HSCs during alcoholic liver injury [21].

Anti-miRNA-21 treatment was also found to prevent the binding of miRNA-21 to the 3′-untranslated region of the VHL mRNA, demonstrating that VHL is a direct target of miRNA-21 [21]. To further advance the mechanistic insights into the role of miRNA-21, RT-PCR and immunofluorescence revealed that miRNA-21 depletion blocked NF-κB activation in human HSCs, both in cultured HSCs and in HSCs isolated from the livers of mice with alcohol-related liver disease, using laser capture microdissection [21]. Collectively, this study suggested that miRNA-21 depletion may reduce pro-inflammatory cytokine production in HSCs and impair macrophage chemotaxis during alcoholic hepatitis.

### 3.5. miRNA-148a

Just as alcohol exposure has been shown to upregulate certain miRNAs, several miRNAs are downregulated and have been scarcely explored. One of these miRNAs, miRNA-148a, is abundant in hepatocytes and has already been shown to regulate hepatocyte differentiation [22]. A 2018 study verified through quantitative reverse transcription PCR that miRNA-148a levels were markedly decreased, both in the liver samples of patients with alcoholic hepatitis and also in mice models subjected to the Lieber-DeCarli alcohol diet and binge alcohol drinking methods [23]. To further explore the molecular mechanism involved in the dysregulation of miRNA-148a, Heo et al. utilized a gene ontology analysis followed by qRT-PCR assay verification to identify FOXO1 as a significantly repressed gene in human alcoholic hepatitis liver samples and mice subjected to binge alcohol drinking [23]. This gene encodes the forkhead box protein O1 (FoxO1) transcription factor, which was previously known to help regulate gluconeogenesis and glycogenolysis through insulin signaling and play a central role in the adipogenesis of preadipocytes [24]. Subsequent correlation analyses corroborated a positive correlation between FOXO1 mRNA levels and miRNA-148a levels in human alcoholic hepatitis samples (r = 0.617, *n* = 13, *p* = 0.008) as well as in mouse liver samples, with the binge alcohol samples exhibiting an even stronger correlation (r = 0.721, *n* = 7, *p* = 0.004) [18]. To validate the effect of FoxO1 as it relates to miRNA-148a levels, the authors used a chromatin immunoprecipitation (ChIP) assay to confirm that FoxO1 specifically bound to the promoter region of the human *MIRNA148A* gene [23].

The authors further identified that thioredoxin-interacting protein (TXNIP) levels are overexpressed in primary hepatocytes treated with ethanol [23]. TXNIP acts as a negative regulator of the thioredoxin redox system, inhibiting thioredoxin’s antioxidant function and serving to regulate various cellular processes, including oxidative stress, apoptosis, and inflammation [25]. Specifically, it activates the NOD-like receptor pyrin domain-containing 3 (NLRP3) inflammasome, a multiprotein complex that cleaves pro-caspase-1 to active caspase-1, which then mediates the propagation of pro-inflammatory cytokines [26]. Through gene analysis corroborated by flow cytometric analysis, Heo et al. were able to conclude that exposure of hepatocytes to alcohol induces TXNIP overexpression through dysregulation of the FoxO1-mediated miRNA-148a expression [23] (see Figure 5). Furthermore, it was shown that alcohol treatment elicited caspase-1-mediated pyroptosis (a lytic form of programmed cell death characterized by cellular swelling, plasma membrane rupture, and release of inflammatory cytokines) through TXNIP overexpression, a process that was shown to be reversed by miRNA-148a transfection [23]. These findings underscore the significance of miRNA-148a in diminishing alcohol-induced inflammasome activation and pyroptosis through TXNIP inhibition [23].

### 3.6. miRNA-30e

Xin et al. established models of alcoholic fat infiltration (AFI) and alcohol-related hepatitis in mice by again treating them with the Lieber-DeCarli alcohol diet and observing a downregulation in miRNA-30e levels in alcohol-related hepatitis [27]. Furthermore, it was discovered that a stepwise decrease in miRNA-30e levels correlated strongly with an increase in Uncoupling protein 2 (UPC2) as AFI progressed into more advanced stages of alcoholic liver disease, including alcoholic hepatitis [27]. This suggested an inverse relationship between miRNA-30e and UPC2 levels. Additionally, the decreased miRNA-30e levels correlated with a decrease in ATP and H_2_O_2_ levels [27]. Because UCP2 is known to uncouple mitochondrial oxidative phosphorylation (thereby reducing ATP production and reactive oxygen species generation), Xin et al. argue that miRNA-30e restores mitochondrial energy metabolism and elevates oxidative metabolism by repressing UPC2 (as reflected by H_2_O_2_), which may paradoxically alleviate the pathological inflammation observed in AH.

This implied that the therapeutic effect of miRNA-30e as it relates to alcoholic hepatitis resided in influencing oxidative stress and energy metabolism [27]. The decline of miRNA-30e in ALD may permit overexpression of UCP2, impairing mitochondrial energy and redox balance [27]. Restoring miRNA-30e ameliorates liver injury, likely by re-balancing ATP generation and oxidative stress through suppression of UCP2.

## 4. Circulating miRNAs

Complex gene regulatory networks in normal and disease-specific physiological conditions are, to some degree, mediated by miRNA-miRNA crosstalk [28]. This crosstalk itself can be altered in specific disease states, including alcoholic hepatitis, such that the inflammatory cascade becomes dysregulated [29].

### 4.1. miRNA-27a

Saha et al. explored the effect of alcohol on crosstalk between human monocytes as it related to extracellular vesicle (EV) production and the involved miRNA. EVs act as vehicles of cell–cell signaling and contain bioactive molecules, including proteins and miRNA [30]. To evaluate the EV secretion pattern of monocytes, the authors subjected human blood monocytes to varying concentrations of ethanol. They measured the EV concentration using Nanosight instrumentation, discovering that the total number of EVs produced by these monocytes increased in a dose-dependent manner [31]. In the same study, the investigators measured the expression levels of macrophage-specific markers on EV-treated monocytes. They concluded that there was a significant increase in the frequency of CD206-expressing and CD163-expressing monocytes, suggesting that EVs derived from ethanol-treated monocytes stimulate naïve cells to differentiate into M2 macrophages [31]. After profiling the miRNA expression levels within EVs derived from alcohol-treated monocytes, it was discovered that there was a significantly high concentration of miRNA-27a in them [31]. Given this selective increase in miRNA-27a, the functional role of this miRNA was next examined by incubating naïve human monocytes with EVs loaded with miRNA-27a mimics. This arm of the study revealed an increase in the M2 macrophage markers CD206 and CD163, indicating that miRNA-27a in EVs is associated with the differentiation of naïve monocytes into M2 macrophages in a dose-dependent manner [31].

### 4.2. miRNA-181

A separate study conducted by Eguchi et al. aimed to identify specific genes that may be targeted by the miRNA residing in the EVs produced by alcoholic hepatitis-associated hepatocytes. This study was based on previous knowledge that nuclear receptor subfamily 1 group D member 2 (Nr1d2) was a specific mRNA that was upregulated in quiescent HSCs based on whole genome microarray [32]. Thus this gene seemed to be a presumed target of not just miRNA-27a but also miRNA-181 which was also known to be upregulated in encapsulated EVs from alcoholic hepatitis-derived hepatocytes [33]. Eguchi et al. tested this notion by examining *Nr1d2* expression in primary HSCs transfected with miRNA-27a and miRNA-181 and discovered that this dose-dependently reduced NR1d2 mRNA levels and inhibited Nr1D2 protein expression [33]. This suggested that the EVs derived from alcoholic hepatitis-associated hepatocytes contribute to HSC activation via miRNA-27a and miRNA-181 targeting and repressing the quiescent HSC genes [33].

### 4.3. miRNA-122, miRNA-30a, and miRNA-192

Alanine aminotransferase (ALT) and aspartate aminotransferase (AST) levels are commonly used to detect liver damage, but they lack specificity. Elevated ALT/AST levels do not indicate the type of liver disease, presence of inflammation, or accurately reflect the severity or stage of liver injury or fibrosis [34]. Additionally, their elevation is not unique to alcoholic hepatitis and occurs in various liver conditions [34]. Knowing that EVs/exosomes are an enriched source of miRNAs and can serve as stable reservoirs of biomarkers in various diseases, Momen-Heravi et al. set out to investigate whether circulating exosomes and their miRNA signatures could serve as non-invasive biomarkers of alcoholic hepatitis [35]. They first discovered that the total number of circulating EVs/exosomes was significantly increased in the sera of alcohol-fed mice compared to controls (*p* < 0.005) [35]. This was accomplished by subjecting the experimental arm of the mice subjects to a 4-week period of the Lieber–DeCarli diet followed by exosome isolation using a combination of serial filtration and precipitation reagent called ExoQuick [35]. This finding suggested that exosome release mirrors ethanol-induced liver injury to a certain extent. Of the list of deregulated miRNAs in EVs isolated from alcohol-fed mice and control mice, confirmatory microarray and qPCR analysis revealed that miRNA-192, miRNA-122, and miRNA-30a were significantly elevated in alcohol-fed mice [35]. In line with the marked upregulation of these miRNAs, they also demonstrated strong potential as diagnostic biomarkers of alcoholic hepatitis. To evaluate their diagnostic performance, Receiver Operating Characteristic (ROC) curve analysis was conducted. Among them, miRNA-192 exhibited the highest area under the curve (AUC = 0.96; *p* < 0.001), followed by miRNA-122 (AUC = 0.92; *p* < 0.001) and miRNA-30a (AUC = 0.85; *p* < 0.05) [35].

The study also analyzed samples obtained from 14 confirmed alcoholic hepatitis patients and 10 healthy controls. Using microarray analysis, Momen-Heravi et al. discovered that levels of miRNA-30a and miRNA-192 were significantly elevated in exosomes from patients with alcoholic hepatitis compared to healthy controls [35]. While miRNA-122 levels were also higher in alcoholic hepatitis patients, the increase was not statistically significant, possibly due to differences in disease stage or model limitations [35]. Importantly, miRNA-192 was consistently upregulated in both mouse models and human samples, showing strong diagnostic potential based on ROC curve analysis, with an AUC of 0.95 (*p* < 0.001) in the alcoholic hepatitis patient samples [35]. miRNA-192 itself is a liver-enriched miRNA not significantly altered in other liver diseases, such as viral hepatitis or hepatocellular carcinoma [36]. Interactome analysis further revealed that miRNA-192 may influence TGF-β/Smad and Jak2/Arhgef1 signaling pathways, which are heavily involved in alcoholic hepatitis pathogenesis [37]. miRNA-30a, expressed in heart tissue, also increased with alcohol exposure, suggesting systemic effects of alcohol beyond the liver [38]. This postulation is in tandem with the fact that chronic alcohol administration is strongly correlated with cardiomyopathies, both in humans and animal models [38]. The heterogenous origins of exosome-associated miRNAs like miRNA-30a suggest that circulating miRNAs, particularly when combined with complementary biomarker profiles, may provide a more powerful diagnostic window into the systemic pathophysiological effects of alcohol.

## 5. Mechanisms of Alcoholic Hepatitis

A myriad of complex mechanisms underscore the onset and progression of alcoholic hepatitis, including the formation of Mallory–Denk bodies and the inhibition of hepatocellular regeneration. Several studies have observed key changes in the expression levels of specific miRNAs as they relate to these unique mechanisms.

### 5.1. miRNA-34a and miRNA-483-3p

Mallory–Denk bodies (MDBs) are cytoplasmic hyaline aggregates found in hepatocytes and serve as a hallmark histological feature of alcoholic hepatitis [39]. They are thought to be formed due to the cytoskeletal collapse of hepatocytes in direct response to the toxic effects of alcohol and other metabolites in alcoholic hepatitis [40]. MDBs primarily consist of p62 (a sequestosome), ubiquitin, and intermediate filament proteins, including keratin 8 (K8) and keratin 18 (K18). Consequently, the ratio of K8 to K18 is generally increased during the formation of MDBs, as K8 tends to be insoluble and resistant to proteasomal degradation [39]. While MDBs are commonly observed in alcohol-related hepatitis, their molecular regulation has been poorly understood. Liu et al. investigated two dysregulated miRNAs, miRNA-34a and miRNA-483-3p, in the context of alcoholic hepatitis and MDB formation and aimed to underscore their mechanistic roles in alcoholic hepatitis pathogenesis. The authors utilized human liver biopsies from patients with alcoholic hepatitis and histologically normal controls. They also used a mouse model of MDB formation in which mice were fed a diet containing 3,5-diethoxycarbonyl-1,4-dihydrocollidine (DDC) to induce oxidative stress and hepatocyte injury [41]. The study demonstrated a significant increase in miRNA-34a expression in liver samples from patients with alcoholic hepatitis and MDBs, as well as in DDC-fed mice [41]. miRNA-34a is known to be regulated by p53 as the miRNA-34 gene promoters contain p53-binding sites [42]. Interestingly, despite the observed upregulation of miRNA-34a in this study, p53 levels were downregulated in both human alcoholic hepatitis and DDC mouse livers [41]. This suggests that alternative, p53-independent pathways, such as chronic oxidative stress and inflammatory cytokines, may be driving miRNA-34a expression in the setting of alcoholic liver injury [41]. Future research would be warranted to clarify the transcriptional control of miRNA-34a in the absence of functional p53.

Conversely, miRNA-483-3p levels were markedly reduced in human alcoholic hepatitis and the MDB-positive mouse model [41]. miRNA-483-3p is a tumor suppressor exerting anti-proliferative properties in response to cellular injury [43]. It does so by targeting CDC25A phosphatase, which prevents its association with cyclin D and blocks cells in the early G1 phase of the cell cycle, suggesting an important role of miRNA-483-3p in cell cycle arrest [43]. Loss of miRNA-483-3p expression could therefore enable uncontrolled cell growth and fibrogenic signaling, further worsening liver injury and increasing susceptibility to neoplastic transformation [41]. Both miRNA-34a and miRNA-483-3p showed strong correlations with the histological presence of MDBs. Increased miRNA-34a and decreased miRNA-483-3p were observed in hepatocytes containing MDBs, suggesting a causal or contributory role in their formation [41].

### 5.2. Role of Ethanol

Although miRNA is a major factor in the pathogenesis of alcohol-related hepatitis, it is important to acknowledge the role that ethanol itself plays. Ethanol is a central and direct pathogenic driver of alcohol-related hepatitis, exerting toxic effects on hepatocytes independent of secondary inflammatory mediators. Following ingestion, ethanol is primarily metabolized in hepatocytes by alcohol dehydrogenase and CYP2E1, generating acetaldehyde and reactive oxygen species (ROS). Both metabolites are highly toxic and promote lipid peroxidation, mitochondrial dysfunction, and DNA damage [44]. Acetaldehyde forms stable protein adducts that impair cytoskeletal integrity and disrupt intracellular trafficking, contributing to hepatocyte ballooning and Mallory–Denk body formation [44]. Simultaneously, CYP2E1 induction amplifies oxidative stress and sensitizes hepatocytes to inflammatory cytokines such as tumor necrosis factor-α (TNF-α), thereby lowering the threshold for cell death in the setting of ongoing alcohol exposure [44].

Beyond its direct hepatotoxicity, ethanol also acts as a potent immunomodulatory stimulus that amplifies hepatic inflammation through gut–liver axis dysfunction. Chronic ethanol consumption disrupts intestinal tight junction integrity, increasing portal translocation of LPS, which activates TLR4 signaling in Kupffer cells and hepatic stellate cells [45]. This results in robust activation of NF-κB–dependent cytokine programs, including TNF-α, IL-1β, and IL-6, which further propagate hepatocellular injury and neutrophilic infiltration characteristic of AH [45].

A full list of this manuscript’s cited dysregulated miRNAs (including their function, involved pathways, expression in AH, strength of existing evidence, and their prospective clinical relevance) is provided in Table 1 below.

## 6. Conclusions

In conclusion, microRNAs play a multifaceted and central role in the pathogenesis of alcohol-related hepatitis by seemingly modulating inflammation, immune responses, hepatocyte injury, and fibrogenesis. Several miRNAs are either upregulated (e.g., miR-217, miR-182, let-7b, miR-21, miR-34a) or downregulated (e.g., miR-148a, miR-30e, miR-483-3p) in alcohol-related hepatitis, each influencing distinct but interconnected pathways. These miRNAs act through various mechanisms, including NF-κB activation, inflammasome signaling, ductular reaction, bile acid metabolism, and regulation of oxidative stress. Moreover, the discovery of miRNAs in circulating extracellular vesicles highlights their potential as non-invasive biomarkers for diagnosis and prognosis. Continued research into miRNA-targeted therapies may open new avenues for the treatment of alcoholic hepatitis, a disease with limited current options and high mortality.

Among the miRNAs reviewed, several emerge as high-priority candidates for further translational investigation based on the strength of mechanistic evidence, availability of human validation, and clinical relevance. miRNA-192 currently represents the most promising diagnostic biomarker, demonstrating consistently high diagnostic performance across both animal models and human cohorts, with relative specificity for hepatic injury. miRNA-182 stands out as a strong prognostic biomarker, given its robust correlation with MELD score, bilirubin, ductular reaction, and short-term mortality. From a therapeutic standpoint, miRNA-148a and miRNA-30e represent particularly compelling targets due to their direct roles in suppressing inflammasome activation, pyroptosis, and mitochondrial dysfunction—central pathogenic mechanisms in alcohol-related hepatitis. In contrast, miRNAs such as let-7b, miR-21, and miR-27a, while mechanistically insightful, currently lack sufficient human validation to justify immediate clinical translation and should be considered second-tier candidates.

From a biomarker development perspective, future studies must prioritize large, multicenter, prospective cohorts with careful phenotypic stratification of alcohol-related hepatitis severity, standardized miRNA quantification platforms, and longitudinal sampling to assess diagnostic accuracy, prognostic performance, and response to therapy. Multimodal biomarker panels that integrate miRNAs with conventional laboratory markers and inflammatory cytokine profiles are likely to outperform single-marker approaches.

From a therapeutic development standpoint, miRNA-based interventions such as miRNA mimics hold theoretical promise but face substantial hurdles, including targeted hepatic delivery, off-target effects, immune activation, and long-term safety. Preclinical work using liver-specific delivery systems, such as lipid nanoparticles or engineered extracellular vesicles, represents a critical next step.

The strengths of this review lie in its comprehensive scope (covering multiple relevant miRNAs involved in alcohol-related hepatitis as well as key mechanistic pathways) and a good selection of foundational primary literature covering evidence from mechanistic studies, animal models, and human studies. The authors also highlight potential therapeutic biomarkers that may hold potential for further drug development opportunities. And by focusing on miRNAs, the review overall explores an evolving area of hepatology research that links epigenetics, inflammation, and clinical outcomes.

However this review is not without its limitations. Certain conceptual and scientific limitations remain, including oversimplification of causality (some studies associating miRNA with AH may be mechanistically correct in controlled experiments but may not be the case when extended to human models) and limited integration of miRNA findings to clinical outcome predictors (MELD, Lille score, etc.). Furthermore, many of these cited miRNA profiles are not completely specific to alcohol-related hepatitis as they are also dysregulated in other hepatic disease processes including NAFLD, viral/autoimmune hepatitis, PBC, PSC, and HCC. This review also cites a relatively small number of human cohort studies. These select studies often lack adjustment for confounders including coexisting metabolic liver disease, viral hepatitis, etc. While this manuscript highlights the discriminatory nature of miRNAs in the setting of alcohol-related hepatitis, they are largely supported by only in vitro or animal studies. Further investigations are warranted in human studies to corroborate clinical applicability. The heterogeneity across all these cited studies, including differences in sample preparation and RNA extraction kits, further weakens interpretive reliability.

Overall this review provides an excellent synthesis of how miRNAs might influence the pathogenesis of alcohol-related hepatitis, but methodological shortcomings limit its applicability, particularly when extrapolating to clinical practice.

## Figures and Tables

**Figure 1 cimb-47-01048-f001:**
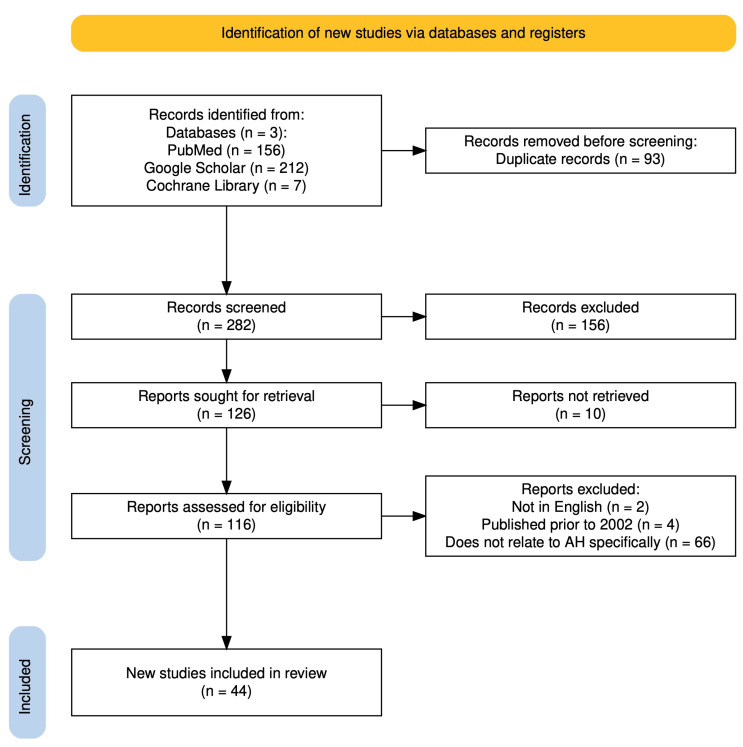
PRISMA flow diagram. Created on Shiny App software v1 [9].

**Figure 2 cimb-47-01048-f002:**
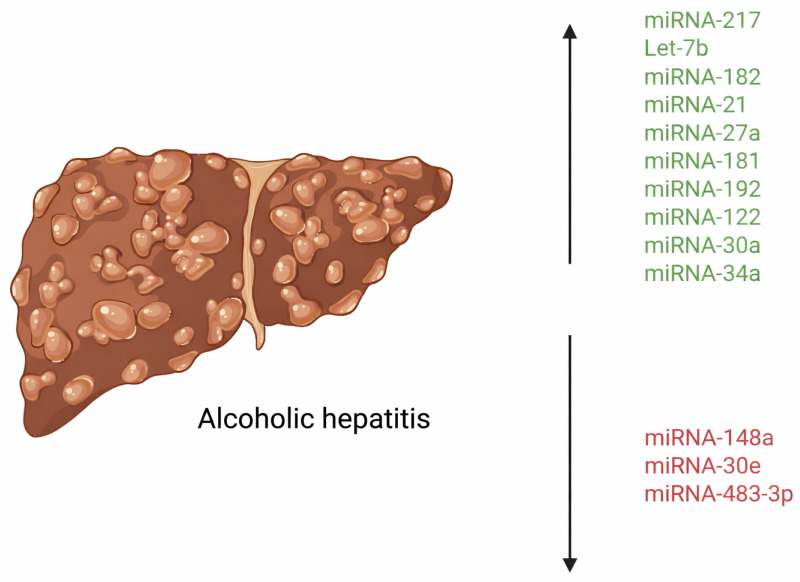
Deregulated miRNAs in alcoholic hepatitis. Created at https://BioRender.com (accessed 24 August 2025).

**Figure 3 cimb-47-01048-f003:**
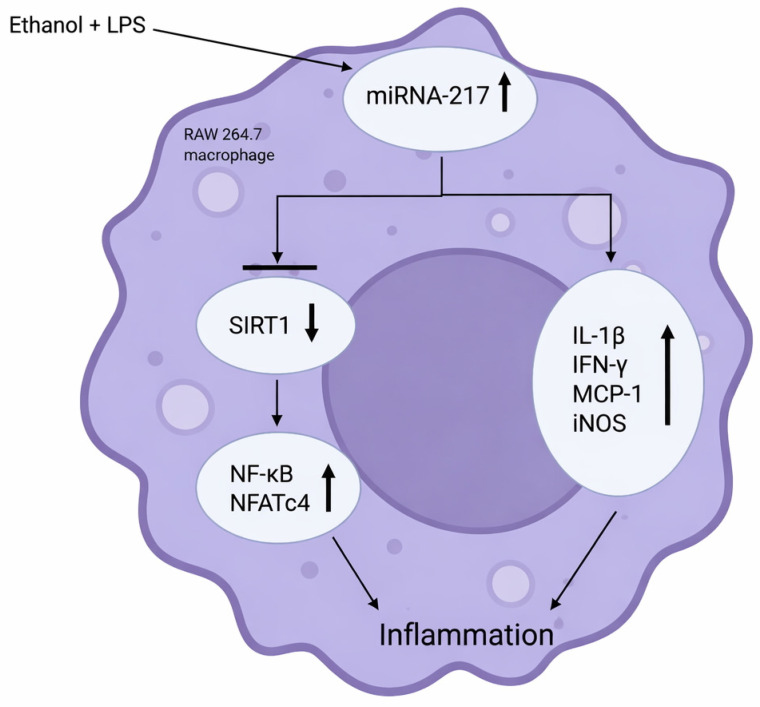
Role of miRNA-217 in ethanol + LPS-induced inflammation in in vitro RAW 264.7 macrophages. Created in https://BioRender.com (accessed 24 August 2025).

**Figure 4 cimb-47-01048-f004:**
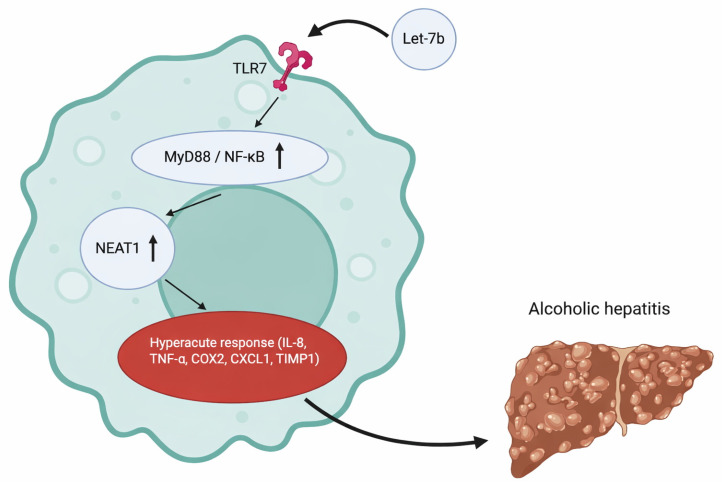
Activation of TLR7 receptors by let-7b leads to downstream pro-inflammatory process which contributes to the onset of alcoholic hepatitis. Created at https://BioRender.com (accessed 24 August 2025).

**Figure 5 cimb-47-01048-f005:**
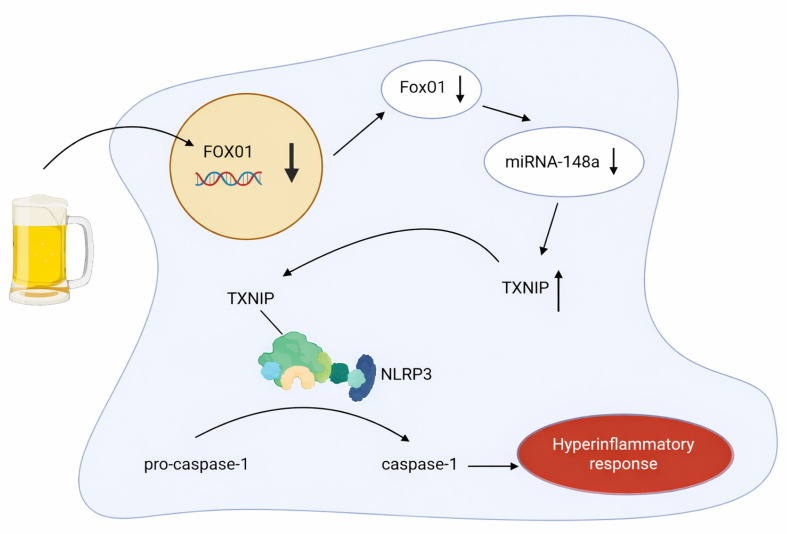
Model for TXNIP-mediated pyroptosis in hepatocytes due to inhibited FOXO1 mRNA transcription from alcohol consumption. Created at https://BioRender.com (accessed 24 August 2025).

**Table 1 cimb-47-01048-t001:** Dysregulated miRNAs in alcoholic hepatitis and their mechanistic and clinical relevance [5]. Strength of evidence is graded qualitatively based on human data availability, experimental validation, and consistency across studies.

miRNA	Principal Function	Key Target(s)/Pathway(s)	Model Type	Expression in AH	Strength of Evidence	Translational Significance
miRNA-217	Pro-inflammatory signaling	SIRT1 → NF-κB	Animal + in vitro mechanistic	Up	Moderate	Mechanistically strong; limited human validation
Let-7b	Innate immune activation	TLR7 → MyD88/NF-κB	Mouse, human RNA-seq	Up	Low	Links ethanol exposure to TLR-driven inflammation
miRNA-182	Inflammation, ductular reaction	Mcp-1, Ccl20, Cxcl5, Cxcl1, Bcl2	Human liver, animal	Up	Moderate	Correlates with MELD and bilirubin; limited replication in large human cohorts
miRNA-21	Stellate cell activation, inflammation	VHL → NF-κB	Animal, HSC mechanistic	Up	Low	Good mechanistic data; weak clinical correlation
miRNA-27a	M2 macrophage polarization	CD206, CD163	In vitro, animal EV	Up	Low	Early mechanistic stage; no human data
miRNA-181	Hepatic stellate cell activation	HSC genes	In vitro, animal EV	Up	Low	Early mechanistic stage; no human data
miRNA-192	Exosomal signaling, fibrosis	TGF-β/Smad, Jak2/Arhgef1 signaling pathways	Human plasma, animal	Up	Moderate	Promising diagnostic marker; high translational directness
miRNA-122	Hepatocyte injury, lipid metabolism	CD320, AldoA, BCKDK	Human, animal ALD	Up	Moderate	Sensitive liver injury marker; not AH-specific
miRNA-30a	Autophagy inhibition	Beclin-1	Human, animal	Up	Moderate	Consistent directionality; small cohorts
miRNA-34a	Mallory–Denk Body formation	NAMPT, SIRT1	Human, animal	Up	Low	Not AH-specific; elevated in multiple liver diseases
miRNA-148a	Pyroptosis suppression	TXNIP → NLRP3	Human, mouse	Down	Moderate	Mechanistic and translationally coherent; small human sample
miRNA-30e	Regulation of ROS	ATP, H_2_O_2_	Human, animal	Down	Moderate	Small cohorts; consistent directionality
miRNA-483-3p	Tumor suppressor	CDC25A phosphatase	Human, mouse	Down	Low	Very limited studies; possible link to fibrosis and neoplasia

Abbreviations: AH, alcohol-related hepatitis; HSC, hepatic stellate cell; NF-κB, nuclear factor kappa-light-chain-enhancer of activated B cells; TLR, Toll-like receptor; MELD, model for end-stage liver disease.

## Data Availability

No new data were created or analyzed in this study. Data sharing is not applicable to this article.

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
