# Peer review of "miRNAs and Alcohol-Related Hepatitis"

_cimb, 2025, doi:10.3390/cimb47121048_

Round 1

Reviewer 1 Report

Comments and Suggestions for Authors

The submitted manuscript is focused on different miRNAs observed in alcohol-related hepatitis. The results of searching in the Scopus database, just using two keywords “miRNA” and “alcohol-related hepatitis”, provide 6 records. The subject has been very poorly described, and at present, it is very limited amount of data worth revising. The Authors did not provide the keywords which were used in literature searching - just "controlled vocabulary terms". There is no novelty to link miRNA with proinflammatory activity. The figures are poor and rather general. In addition, the table falls short of the standards set for tables in scientific papers.

In addition, there are some reviews in this field, e.g.

Pathology Research and Practice, 243, 154375, 2023

Clinics and Research in Hepatology and Gastroenterology, 41(3), pp. 254–261, 2017

Pointing out miRNA as a factor involved in the pathogenesis of alcohol-related hepatitis is important, but the first factor related to it is ethanol itself.

Author Response

Comment 1: "The Authors did not provide the keywords which were used in literature searching - just "controlled vocabulary terms"."

Response 1: Thank you for pointing this out. I expanded the keyword phrasing to include additional keywords which were used in the original literature search as it pertains to more specific pathways or mechanistic details of AH pathogenesis. Please see revised keywords section on page 1.

Comment 2: "The table falls short of the standards set for tables in scientific papers"

Response 2: Agree, the table certainly needed more refining to better highlight the main points of each miRNA. I re-did the table and separated mechanism vs. clinical utility, removed redundancy and vague phrasing, provided more precise molecular targets, and added stronger translational framing for clinical applicability purposes.

Comment 3: "Pointing out miRNA as a factor involved in the pathogenesis of alcohol-related hepatitis is important, but the first factor related to it is ethanol itself."

Response 3: This was certainly a key point that was missed in the earlier draft. I added a new section towards the end of page 10 that highlights how ethanol plays a role in AH pathogenesis.

Reviewer 2 Report

Comments and Suggestions for Authors

General assessment:

The manuscript provides a broad overview of microRNA involvement in alcohol-related hepatitis (AH). While it covers many relevant studies, the overall presentation lacks critical depth, methodological rigor, and translational insight expected of a high-quality review. Several conceptual, structural, and scientific issues should be addressed before the manuscript can be considered for publication.

Specific comments:

  1. Lack of clarity regarding the review type

Although a “Search Methodology” section is provided, the manuscript does not clearly indicate whether it is intended as a narrative review or a systematic/semi-systematic review.

No PRISMA diagram is provided.

No detailed inclusion/exclusion criteria.

Search strategy lacks transparency regarding filters, study selection, and data extraction.

This raises concerns about selection bias and limits the credibility of the literature synthesis.

  1. Uneven depth of discussion across miRNAs

Some miRNAs (e.g., miR-217, miR-182, miR-21) are discussed in detail, while others (miR-30e, miR-483-3p, miR-30a) are described only briefly and superficially.

  1. Lack of integrated mechanistic interpretation

Although individual pathways—NF-κB activation, inflammasome signaling, ductular reaction, oxidative stress—are mentioned, the review does not integrate them into a coherent mechanistic framework or present an overarching miRNA regulatory network in AH.

Without such integrative analysis, the review remains descriptive rather than analytical.

  1. Overinterpretation of preclinical data

Several miRNAs highlighted as potential biomarkers or therapeutic targets are supported mainly by in vitro or small animal model studies.

For example, miR-182, miR-21, and miR-148a have limited human validation.

Yet the manuscript describes them in a manner that overstates their clinical applicability.

  1. Lack of disease specificity analysis

Many miRNAs discussed (e.g., miR-122, miR-34a) are not specific to alcoholic hepatitis, and are also elevated in viral hepatitis, NAFLD, or HCC.

The manuscript does not appropriately address these confounding factors.

Claims regarding diagnostic or prognostic relevance should be toned down and contextualized.

  1. Therapeutic potential is discussed superficially

The manuscript mentions the potential of miR-148a, miR-21, and others as therapeutic targets, but fails to discuss: challenges in miRNA delivery, off-target effects, tissue specificity, long-term safety, barriers to translation into clinical trials

  1. Limitations of EV/exosome-based biomarkers not addressed

Although circulating exosomal miRNAs are highlighted as promising biomarkers, the paper lacks discussion of: variability in EV isolation techniques, lack of clinical standardization, pre-analytical variables, inconsistencies across studies

Without a balanced analysis, the translational value is overstated.

  1. Figures are conceptual rather than analytical

Most figures are simplified BioRender-style illustrations that do not contribute new mechanistic insight or summarize comparative data.

  1. Evidence grading in Table 1 lacks transparency

Table 1 categorizes evidence strength (Low/Moderate) but does not specify: criteria used, weighting of human vs. animal vs. mechanistic data, reproducibility, sample sizes

This weakens the table’s scientific reliability.

  1. Limitations section and conclusion are too general

The conclusion reiterates general observations (e.g., miRNAs play multifaceted roles) but does not provide: clear prioritization among miRNAs, critical gaps in current knowledge, methodological improvements needed in future studies, a roadmap toward clinical application (biomarker validation, therapeutic development)

As a review, it should offer stronger critical insight and forward-looking guidance.

  1. Potential citation bias

The manuscript heavily relies on a limited group of authors and studies (e.g., Blaya, Momen-Heravi, Heo), without incorporating: broader AH multi-omics literature, recent (2023–2025) large-scale human studies, more diverse international research groups

This limits the comprehensiveness and balance of the review.

Author Response

Comment 1: Lack of clarity regarding the review type.
Response 1: Thank you for highlighting this. In the Search Methodology section on page 2, I start the section by mentioning that this manuscript is indeed a systematic review. I also added a PRISMA diagram (Figure 5) that underscores the search methodology used for this review. I included inclusion and exclusion criteria on this figure as well.

Comment 2: Uneven depth of discussion across miRNAs

Response 2: We agree with this critique. I further elaborated on miRNA 30e in section 3.6 on page 7 as well as miRNA-30a in section 4.3.

Comment 3: Overinterpretation of preclinical data

Response 3: Thank you. We added a disclaimer in the Conclusion section about how these potential biomarkers are largely supported by in vitro or animal models, and that further human studies would help corroborate the clinical utility of these biomarkers.

Comment 4: "Many miRNAs discussed (e.g., miR-122, miR-34a) are not specific to alcoholic hepatitis, and are also elevated in viral hepatitis, NAFLD, or HCC. The manuscript does not appropriately address these confounding factors."

Response 4: We added to the conclusion section on page 13 about how many of the miRNAs discussed are not just specific to alcoholic hepatitis but to other pathologies as well.

Comment 5: "Therapeutic potential is discussed superficially. The manuscript mentions the potential of miR-148a, miR-21, and others as therapeutic targets, but fails to discuss: challenges in miRNA delivery, off-target effects, tissue specificity, long-term safety, barriers to translation into clinical trials."

Response 5: This was a good point that we are now addressing in the Conclusion section. On page 13 in paragraph 2 ("From a therapeutic development standpoint...") we address the avenues of future research including liver-specific delivery systems.

Comment 6: "Evidence grading in Table 1 lacks transparency. Table 1 categorizes evidence strength (Low/Moderate) but does not specify: criteria used, weighting of human vs. animal vs. mechanistic data, reproducibility, sample sizes. This weakens the table’s scientific reliability."

Response 6: The table certainly needed some improvement. We revised the table to separate mechanism vs. clinical utility, removed redundancy and vague phrasing, gave more precise molecular targets, and stronger translational framing for clinical utility sake.

Comment 7: Limitations section and conclusion are too general. The conclusion reiterates general observations (e.g., miRNAs play multifaceted roles) but does not provide: clear prioritization among miRNAs, critical gaps in current knowledge, methodological improvements needed in future studies, a roadmap toward clinical application (biomarker validation, therapeutic development). As a review, it should offer stronger critical insight and forward-looking guidance."

Response 7: Conclusion section was revised to provide a clearer understanding of which miRNAs seem promising with regards to further research, and also highlighted critical gaps in knowledge. In the first and second paragraphs on page 13 there are now two new paragraphs that highlight biomarker validation and therapeutic development.

Reviewer 3 Report

Comments and Suggestions for Authors

This review article is engaging and provides important insights into microRNA-associated molecular mechanisms underlying alcohol-related hepatitis. The authors have also sufficiently discussed the study’s limitations. However, a few points require clarification, and some typographical errors should be corrected.

1. Abstract, lines 4–6 from the bottom: The following sentence is unclear in its current form: “Considering miRNAs are pivotal regulators of AH pathophysiology, modulating immune activation, hepatocyte death, fibrosis, and metabolic dysregulation.” Please revise this sentence for grammatical clarity and logical flow.

2. Typographical errors: Some typographical errors are present throughout the manuscript. Please correct these, including but not limited to: (1) Page 6, line 11; (2) Page 10, second paragraph, line 7; and (3) others.

Author Response

Comment 1: "Abstract, lines 4–6 from the bottom: The following sentence is unclear in its current form: “Considering miRNAs are pivotal regulators of AH pathophysiology, modulating immune activation, hepatocyte death, fibrosis, and metabolic dysregulation.” Please revise this sentence for grammatical clarity and logical flow."

Response 1: Thank you for pointing this out. This particular sentence has been revised for grammatical clarity. Seems that this was a punctuation error that was missed upon submission of the draft.

Comment 2: "Typographical errors: Some typographical errors are present throughout the manuscript. Please correct these, including but not limited to: (1) Page 6, line 11; (2) Page 10, second paragraph, line 7; and (3) others."

Response 2: We scanned the document and made several corrections to some typographical and/or grammatical errors including the two examples you provided.

Reviewer 4 Report

Comments and Suggestions for Authors

Bandara et al. have aimed to present a summary of the current literature status regarding the role of miRNA in alcohol-related hepatitis. Although this review is of interest in the field and journal, there are several issues (as listed below) need to be addressed before it could be accepted for publication in CIMB.

Abstract

The list of upregulated and downregulated miRNAs is too much information and probably they can be grouped by function instead of listing.

The final concluding statement should be rewritten to be more concise.

Introduction

Paragraph 1 – the statement “Around 20-40%...” should specify the evidence level and add recent citations beyond StatPearls.

Paragraph 2 – you can add briefly how endotoxemia is specifically linked to miRNA changes.

Search Methodology

 Paragraph 1 – the database list should follow PRISMA guidelines. Include a PRISMA flow diagram highlighting the number of articles screened, included and excluded.

Paragraph 2 – The section 2.1 following ‘Search methodology’ should begin as a new main section (Section 3).

Section 2.2

Paragraphs 1 – the discussion is highly dependent on Yin et al. (2015) and probably the authors should include newer validation studies. Add a concluding sentence explaining how miR-217 may be targeted therapeutically.

Section 2.3

Paragraphs 1-2 – discussion on whether human biopsies or plasma studies validate this miRNA as a biomarker is lacking. Also, specify the sample size in the referenced RNA-seq study. The term ‘hyperacute phase’ should be clarified or provided with a citation.

Section 2.4

Paragraph 1 – Add information of if miR-182 is elevated in serum or only in liver tissue based on the report by Blaya et al. (2016). Also, the mechanistic rationale should be included for association between miRNA-182 and ductular reaction.

Paragraph 2 – Some limitations should be added, for example species differences in bile acid metabolism.

Section 2.5

Paragraph 1 – add some information on specificity, that is, if miRNA-21 elevated in other liver diseases.

Section 2.6

Paragraph 1 – explain the difference between pyroptosis and apoptosis/necrosis in AH. The correlation values should be accompanied by sample size (n=?). Indicate if miR-148a has diagnostic value (as it is currently projected as mechanistic).

Section 2.7

Paragraph 1 – this section needs to be expanded on how UCP2-mediated mitochondrial uncoupling contributes to AH and if miRNA is measurable in serum.

Section 2.8

Paragraph 1 – explain how crosstalk influences disease progression with examples.

Section 2-8

Paragraph 1- emphasis the relevance of how does anti-inflammatory M2 macrophages fit with AH pathogenesis.

Section 2.10

 Paragraph 1 – some more in-depth discussion on Nr1d2 role in fibrosis and EV biogenesis alterations in AH.

Section 2.11

Paragraphs 1-3 – it is better to discuss the specificity issues with miR-122 (also elevated in viral hepatitis). Also briefly discuss if combining multiple miRNAs improves diagnostic performance. Add sample size for the human patient cohort. The limitations of EV isolation using ExoQuick known for purity issues can be mentioned.

Section 2.12

Paragraph 1 – Mallory-Denk bodies should be defined.

Section 2.13

Paragraphs 1-3 – the discussion about p53-independent pathways requires citations. Also, explain if these miRNAs can be measured in serum or only in liver tissue. More details on how altered cell cycle regulation contributes to AH progression.

Table 1 – some target genes seem incomplete or too simplified. Also, the ‘strength of evidence’ column should be justified by explaining the scoring criteria in methods.

Conclusion

Paragraph 1-2 – it is better to separate scientific conclusion from review limitations. Some future perspective statements involving (1) potential for miRNA therapeutics, (2) need for standardization of EV isolation and (3) required validation in large cohorts.

Author Response

Thank you for your suggestions. We went through each of your highlighted issues, critically assessed each point that was made, and made appropriate changes to the manuscript. Notably, the final sentence of the introduction is more concisely written, a PRISMA flow diagram highlighting the search methodology and the explored databases and exclusion criteria was created (Figure 5), the "Search Methodology" was revised into a brand new section, specified the sample size in the RNA-seq study mentioned in formerly section 2.3 paragraph 1-2, and highlighted the differences between pyroptosis and apoptosis in formerly section 2.6. We strongly believe that the subject matter itself, now with these changes that you suggested, would greatly contribute to the current literature.

Round 2

Reviewer 4 Report

Comments and Suggestions for Authors

The authors have satisfactorily addressed all the comments raised by reviewers and substantially improved the overall quality of the article. Therefore, I recommend accepting this article for publication in CIMB.